# Eukaryotic Genomes Show Strong Evolutionary Conservation of *k*-mer Composition and Correlation Contributions between Introns and Intergenic Regions

**DOI:** 10.3390/genes12101571

**Published:** 2021-10-01

**Authors:** Aaron Sievers, Liane Sauer, Michael Hausmann, Georg Hildenbrand

**Affiliations:** Kirchhoff-Institute for Physics, Heidelberg University, INF 227, 69117 Heidelberg, Germany; aaron.sievers@kip.uni-heidelberg.de (A.S.); lia.sauer@gmx.de (L.S.); hausmann@kip.uni-heidelberg.de (M.H.)

**Keywords:** *k*-mer, sequence analysis, alignment-free, short tandem repeats (STRs), sequence patterns, eukaryotes, intron, intergenic region, 3D conformation of DNA, evolution, genome

## Abstract

Several strongly conserved DNA sequence patterns in and between introns and intergenic regions (IIRs) consisting of short tandem repeats (STRs) with repeat lengths <3 bp have already been described in the kingdom of *Animalia*. In this work, we expanded the search and analysis of conserved DNA sequence patterns to a wider range of *eukaryotic* genomes. Our aims were to confirm the conservation of these patterns, to support the hypothesis on their functional constraints and/or the identification of unknown patterns. We pairwise compared genomic DNA sequences of genes, exons, CDS, introns and intergenic regions of 34 *Embryophyta* (land plants), 30 *Protista* and 29 *Fungi* using established *k*-mer-based (alignment-free) comparison methods. Additionally, the results were compared with values derived for *Animalia* in former studies. We confirmed strong correlations between the sequence structures of IIRs spanning over the entire domain of *Eukaryotes*. We found that the high correlations within introns, intergenic regions and between the two are a result of conserved abundancies of STRs with repeat units ≤2 bp (e.g., (AT)n). For some sequence patterns and their inverse complementary sequences, we found a violation of equal distribution on complementary DNA strands in a subset of genomes. Looking at mismatches within the identified STR patterns, we found specific preferences for certain nucleotides stable over all four phylogenetic kingdoms. We conclude that all of these conserved patterns between IIRs indicate a shared function of these sequence structures related to STRs.

## 1. Introduction

Genome regions encoding for the chemical structures of proteins, such as genes, exons or CDS (coding DNA sequences), are known to harbor functional sequence structures (amino acid codons) conserved within a wide phylogenetic range [1]. While the remaining "non-coding" regions (introns and intergenic regions (IIRs)) were initially declared as useless “junk” DNA [2,3], the existence and importance of conserved sequence structures in IIRs became clearer and clearer in the last decades [4,5]. Interesting findings include similarities between individual introns [4] as well as conserved global intronic or intergenic sequence structures [5,6]. The first speculations of a functional relation of IIRs in *Animalia* were made in [7], where a correlation between the sizes of the two was observed [7]. More recent studies also found conserved structures comparing different regions, e.g., between introns and intergenic regions [8]. 

While the genomes of the *Animalia* species were analyzed in [8], the aim of this study was to search for conserved sequence structures within IIRs of *Embryophyta*, *Protista* and *Fungi* and to compare the results between the kingdoms (including *Animalia*). Standard sequence analysis tools, such as the NCBI Basic Local Alignment Search Tool (BLAST) [9], cannot be effectively used to search such structures within regions of sizes comparable to entire genomes [10]. Therefore, powerful alignment-free methods were developed and have been established [5,11,12]. We used a simplistic but powerful method called *k*-*mer analysis* [12] designed for this special task [12].

The result of a *k*-mer analysis is a correlation coefficient [13] quantifying the similarity of analyzed sequences. An observed, significantly high correlation within a wide phylogenetic range of organisms indicates sequence structures conserved within that range. While a standard *k*-mer analysis is sufficient to discover the conservation of sequence structures in general, it does not provide information about the sequence patterns responsible for this conservation. Therefore, we additionally analyzed the results by performing *a decomposition of the correlation coefficient* (see [8] for details) to quantify the contribution of individual DNA words to the respective correlation coefficients. The resulting word lists were then analyzed for peculiar patterns. As short tandem repeat (STRs) are dominating the correlations, these repeated sequence patterns were analyzed further to search for evidence of functional and non-random aspects in their occurrence and patterns of distribution in *Embryophyta*, *Protista* and *Fungi*.

## 2. Materials and Methods

### 2.1. Selection of Genome Sequences

To limit selection biases of the analyzed sequences to a minimum, we used complete and unmasked genome sequences with annotated genes, exons and coding CDS. While masking of low complexity regions would reduce computational costs, we used unmasked versions to include STRs and DNA satellites observed to show conserved patterns in *Animalia* [8,12]. Therefore, we downloaded unmasked nucleotide sequences of completely assembled chromosomes from the National Center for Biotechnology Information (NCBI) website (FASTA and GenBank annotations [14]). We decided that the genomes of 93 organisms matching our selection criteria are a sufficiently large and diverse set for producing reliable results. We also added the results for *Animalia*, derived in our previous studies [8]. Since the publication of [8], a number of interesting genomes of *Animalia* taxon were sequenced and annotated with sufficient quality; therefore, we decided to include 18 additional *Animalia* genomes in our analysis. In total, the genome sequences of 149 organisms were analyzed. A complete list of downloaded sequences can be found in the (Appendix A).

### 2.2. k-mer Analysis

We used a method called *k*-mer *analysis* [8] to search and quantify the frequencies of *k*-mer words of different sizes *k* and in different regions (e.g., introns and intergenic regions) of the respective genomes. The resulting *k*-mer spectra were correlated using the *Pearson correlation*. As a reference to test the significance of the results, we also correlated artificial *k*-mer spectra created by zero order *Markov models*, simulating sequences of comparable sizes but with random distributions of nucleotides. We generated five random spectra for each word length *k* sequences with a size of 1000 Mbp each and used the variance between the results for error estimation. Zero order models were used since we did not want to exclude constraints on the G+C content for our null hypothesis at this point. The results of this simulation can be found in the (see Appendix A).

We have to point out that correlations of *k*-mer spectra of different word lengths *k* cannot be directly compared, since spectra of higher word length produce lower correlation values due to the fact that short *k*-mers are rarely unique, while becoming unique given a certain length (see [15] and *k* dependency of results in Appendix A). For that reason, we limited our discussions to comparisons of spectra with the same word length *k*.

### 2.3. Correlation Decomposition

In order to further investigate conserved patterns of different *k*-mer word sets (and individual words), we used a method called *correlation decomposition* [8]. We, again, used the analysis of artificial *k*-mer spectra for reference. We used first order *Markov models* to correct for influences of different G+C contents for this analysis and to exclude the over-representation of certain *k*-mer word families (e.g., tandem repeats) as a trivial result of G+C content. Again, we simulated five independent random spectra for G+C contents from 50% to 25% for *k* = 7 and *k* = 11 with a size of 1000 Mbp each. The results of the simulation can be found in the Appendix (see Appendix A).

## 3. Results

### 3.1. Correlation of k-mer Spectra

We analyzed sequences of different genome regions (CDS, exons, introns and intergenic regions—defined as in [8]) of all organisms listed in Appendix A by using the **k*-mer analysis* described in [8]. Analogous to the parametrization used in [8], we used word lengths *k* ranging from 1 to 11 for the analyses. The mean values of the resulting correlation coefficients within every respective set of regions and genomes were calculated (e.g., the mean correlation coefficient between intronic and intergenic regions of *Animalia*). For reference and also a test of significance, we performed the same analysis on spectra generated by *Markov models* (see Methods for details) and subtracted the results from modeled data from the data derived from real genomes. The results for different work lengths *k* are shown in Figure 1. Moreover, adding 18 additional species to the *Animalia* selection did not change the results of the *Animalia* significantly (Appendix A shows a homogeneous distribution on the genomic level, and Appendix A shows low differences of the mean correlation values between the set of 56 genomes (with 18 new ones) and the set of 38 genome). Therefore, we decided to use the data derived from the set of 38 *Animalia* genomes (as in [2]) for the following analyses instead of repeating the very time intensive calculations.

An illustration of the data as averaged for one data point within Figure 1 is shown as a heatmap for *k* = 7 between individual intronic and intergenic regions in Figure 2. One general result is that, for *Animalia* and *Embryophyta*, all correlations seem to be quite high and homogeneously distributed, while the correlations of *Protista* and of *Fungi* can be divided into two sets. One set is highly correlated with most of the spectra derived from *Animalia* and *Embryophyta* genomes and also with each other (subsets #S1). The other set hardly correlates with any other set including itself (subsets #S2). This is especially true in nearly all correlations calculated (data not shown). As an intuitive approach to finding the appropriate attributions relative to the sets #S1 or #S2, this has been visually performed on the base of genome–genome correlations as shown in Appendix A and as provided in Appendix A. Set #S1 of *Protista* and set #S1 of *Fungi* are also shown in Figure 1 as light curves.

In *Embryophyta,* for *k* < 9, the correlations are highest for intronic–intronic correlations, followed by those of intronic–intergenic and intergenic–intergenic. For higher word length, the values of coding regions show higher correlation coefficients (see Figure 1). For the correlations in *Protista* as well as in *Fungi*, the highest values are mostly obtained for intergenic–intergenic, followed by coding sequence containing elements and reaching or slightly surpassing the mapped intergenic correlations for higher *k*. This is true for the entire sets of these kingdoms as well as for their #S1 subsets, even though the values in these subsets are approaching as high values as it is in *Embryophyta*, although with some of their own characteristics (parallel lines in the graph over nearly all values of *k* for #S1 in *Protista* and a more pronounced slope for higher *k* (*k* > 6) for intronic–intronic and intronic–intergenic correlations in *Fungi*). Intronic–intergenic correlations seem to be close to intronic–intronic ones on a weaker correlation level in these two kingdoms. This result may be related with a still considerable amount of intergenic regions in these kingdoms (still above 30% of genome, about half the value as for *Embryophyta*) but with far lower contents of introns (down to 3% of genome) and higher amounts of genes relative to that (nearly twice as much of genome as in *Embryophyta*) (Table 1).

The results described above and shown in the heatmap in Figure 2 reveal a similar trend for high correlations of spectra of intronic and intergenic regions in *Embryophyta* as already observed in *Animalia* [8]. The high homogeneity and high correlation in the *Embryophyta* can also be observed in detail in Appendix A. This heatmap is provided as an example to show how the correlations may also bridge over taxonomic borders: the eudicotyledons start with *Vitis vinifera*, the big homogeneity seems to start already in the monocotyledons (Liliopsida) with *Ananas comosus*, and also the only moss *Physcomitrella patens* seems to have higher correlations relative to nearly all eudicotyledons than some of the monocotyledons. In some examples, this also extends down to genus as in *Protista* with *Babesia microti* belonging to #S1 and *Babesia bigemina* belonging to #S2.

At first glance, this effect may be related to the G+C content of the genomes analyzed. Nearly all genomes with a genomic G+C content > 45% belong to both #S2 subsets, with a few exceptions > 40%.

Figure 2 also shows a special feature regarding the comparison of the quite homogeneous areas for *Animalia* and *Embryophyta* with less homogeneous areas of *Protista* and *Fungi*. The deep red diagonal indicates a high degree of similarity between IIRs for *Animalia*, *Embryophyta* and *Protista*, regardless of subsets. For *Fungi*, this diagonal is quite weak, pointing to a greater heterogeneity between IIRs in the *Fungi* genomes. Additionally, what is interesting is the fact that the high correlations not only exist within the big kingdoms of *Animalia* and *Embryophyta* together with subsets #S1 in *Protista* and #S1 in *Fungi* but also shows high correlation values between these kingdoms’ IIRs. This observation may indicate a general pattern in these genomic regions conserved in all eukaryotic genomes.

Different features can be observed for *k* = 11 (see Figure 3). The overall correlation values are lower, the high homogeneity between IIRs in the same genome is nearly only kept up in *Animalia* and *Embryophyta,* as can be observed by the diagonal, as well as higher heterogeneity between the genomes even in *Animalia*. The lower correlation value areas also appear in the #S1 subsets of both *Protista* and *Fungi*, illustrating that both subsets in each kingdom gradually differ in terms of homogeneity. Having this in mind, it seems noteworthy to point at the relatively high correlation values between introns of *Animalia* and intergenic regions of *Embryophyta* and especially between introns of *Embryophyta* and intergenic regions in nearly all genomes of *Animalia*, *Embryophyta* and of the subsets #S1 in *Protista* and #S1 in *Fungi*. These values are often far higher than between IIRs within the kingdoms. It seems as that the introns in *Embryophyta* have a strong characteristic shared over many eukaryotic genomes.

### 3.2. Abundance and Correlation Contributions of k-Words

In order to better understand the relevance of certain sets of *k*-mer words for the observed correlations, we performed *k*-mer *correlation contributions* (see Methods) on different word sets. The most contributing words were derived for *k* = 7 and *k* = 11. An example of such lists is provided in Appendix A. As a reference, we calculated *correlation contributions* for spectra generated by *Markov models* (see Methods).

As already reported for *Animalia*, words with structures of STRs dominate these lists for all correlations [8]. When examining the repeat unit length b of these tandem repeats, some special features appear (Table 2 and Table 3): For *k* = 7 in intronic and intergenic, contributions of words representing STRs with repeat unit lengths of *b* = 1 and *b* = 2 and exons and CDS were significantly higher than expected from random spectra for *b* = 3 (see Appendix A for reference). Exonic–intronic correlations seem to be special, as they yield results nearly that are as high for *b* = 1 and *b* = 2 as IIRs correlations and higher values for *b* = 3, especially in in *Protista* as well as in *Fungi*. Many of these characteristics are even stronger for *k* = 11, which is also not expected (see Appendix A). This is also shown in examples in a graphic overview in Figure 4 for *k* = 7 and for *k* = 11. 

While tandem repeat words make up a tiny fraction of all possible *k*-mer words (for *k* = 7: 0.02% if *b* = 1, 0.07% if *b* = 2/for *k* = 11: 0.0001% if *b* = 1, 0.0003% if *b* = 2), they contribute in the IIRs correlations more than 10% for *k* = 7 and more than 20% for *k* = 11 to the actual correlation values. The values derived from simulations (see Methods) were not higher than 2% for *k* = 7 and not more than 0.2% for *k* = 11 (see Appendix A). 

In contrast, contributions found within coding regions can be observed as insignificant. Therefore, tandem repeat words with small repeat unit lengths are mainly responsible for the correlations observed within and between IIRs.

In addition to the fact that tandem repeat words are overrepresented within the top contributing word lists shown, it is also visible that there is a tendency for A/T rich words to be the top contributing words for IIRs and an opposite tendency to be the top contributing words between CDS and exons. To quantify this observation, we also performed a correlation decomposition based on the G+C content of respective *k*-mer words by defining sets such as S_G+C < X%_ = {words with G+C content < X%} (may be defined analogously for A/T). The results are shown in Appendix A, confirming this tendency. In regions with coding sequences, the peak for correlation contributions is found around 25–50% for G+C content, and in IIRs the peak lies in the bin for *k*-words consisting of 100% of A/T bases. 

Since the results for coding regions were insignificant, the focus in the following will be set on the IIRs. On top of this, a pattern in exon and CDS regions based on a repeat unit length of *b* = 3 seems to be quite plausible, as this is consistent with the well-known sequence patterns of amino acid codons that may be responsible for their conservation even over large taxonomic distances. Accordingly, we will focus on repeat unit lengths of *b* = 1 and *b* = 2 in the following.

As an overview for better comparison, the correlation contributions of tandem repeat words in all kingdoms with repeat lengths *b* = 1 and *b* = 2 are given in Figure 5A for *k* = 7 and in Figure 5B for *k* = 11 and in Table 4. For better reading, we will use an extended base code of the following: PolyW is either polyA or polyT and polyS is either polyC or polyG. All other sequences are tandem repeats with a repeat unit length of *b* = 2 with W = {A;T}, S = {C;G}, R = {A;G}, Y = {C;T}, K = {G;T} and M = {A;C}, for example, homoR is ((AG)_n_ and (GA)_n_).

The dominance of A/T words suggested by results of Appendix A is observed in the correlation contributions not only for polyW but also within homoW tandem repeats, with the strongest contributions in *Protista*. For *Embryophyta* as well as *Protista,* when moving from *k* = 7 to *k* = 11, the influence of homoW shown by their correlation contribution exceeds that of polyW by nearly a factor of two, showing that this is not all about polyW stretches. The high contribution of non-polyA and non-polyT words, if taken together, is also again noteworthy. When moving from *k* = 7 to *k* = 11, this becomes even more dominant in all kingdoms, while the influence of tandem repeats increases for all words. In *Animalia,* the rise in all words except for polyS words is remarkably strong. In general, as this is also visible in Figure 5 and 5B, the values of homoR and homoY sequences are in the same range, while the values of homoK and homoM sequences are in their own but also the same range.

For better insights, the special subsets #S1 and #S2 in *Protista* and *Fungi* in this approach have also been examined (Appendix A), showing that what has been observed in *Protista* seems to be strongly influenced by its subset #S2 with its own characteristics and shared with subset #S2 of *Fungi*. These genomes with high genomic G+C content also have strong correlation contributions from STRs but are contrasted with genomes with lower G+C content, as all STR words (with in exception of homoS in subset #S2 of *Fungi*) in the #S1 subsets and in *Embryophyta* and *Animalia* have strong influences, not only polyW and homoW. In #S1 in *Protista* for *k* = 11, polyS is stronger than polyW, and the ones that are by far the strongest are homoK and homoM, showing the characteristics of this subset. In some aspects, #S1 in *Fungi* is still close to low G+C genomes; #S1 in *Protista* seems to be stronger based on polyW and homoW than even *Embryophyta* and *Animalia*. 

It may be interesting that algae, as the assumed closest relatives to *Embryophyta*, are in the special subset of #S2 in *Protista*.

An interesting characteristic feature is given by the values for correlation contributions that are close to each other (an overview of this behavior is given in Table 5). An equal distribution and so an equal correlation contribution for a sequence and inverse complementary sequence on one strand would be considered normal. This is not expected for the value of the sequence with one base shift (e.g., value for ACACACA nearly equal to the value for TGTGTGT but not with the value for CACACAC). Before analysis, some general concepts may be valuable if searching for patterns in inverse complementary and shift sequences. For polyW and polyS, the inverse complementary sequences are in the same subset, and shift sequences are the same as the sequence searched for. For homoW and homoS (and for odd values of *k*), the inverse complementary sequences are in the same subset and are identical with shift sequences. For all other sequences (and for odd values of *k*), the shift sequences are in the same, subset and the inverse complementary sequences are in the complementary subset of bases (homoR is complementary with homoY and homoK with homoM). 

As nearly all findings for “own behavior” are found in *Protista* and *Fungi*, they will be directly analyzed on their subset level. The values for polyW, polyS, homoW and homoS are each nearly always similar for the inverse complementary sequence and follow this pattern in each kingdom, subset and for both *k* analyzed, differing only at one point with their own pattern for *k* = 7 in subset #S2 in *Protista* for polyW and polyS (the value for polyA nearly equal to the value for polyG but not with the value for polyC). It is the same set of sequences in which polyS is stronger than polyW. 

The values for all other sequences (homoR, homoY, homoK and homoM) also behave nearly the same in *Embryophyta* and *Animalia*, with equal values for inverse complementary sequences. For higher *k*, the values for inverse complementary sequences also proceed close to values for shift sequences (labeled ambiguous). In the subsets #S1 and #S2 of *Protista* and *Fungi,* homoK and homoM sequences, but also homoR and homoY, in contrast show a pattern for same values with shift sequences and are even stronger with higher *k* (e.g., value for ACACACA nearly equal to the value for CACACAC but not equal to the value for TGTGTGT). Both subsets #S2 have their own characteristics under this perspective. The subsets of #S1 seem to be between these and *Embryophyta* and *Animalia*.

These patterns are indicated by their special characteristics, and general rules are not the only ones that apply (amount/correlation contribution of sequence and its inverse complementary is the same), but as these rules seem to vary according to subsets of sequences, functional aspects may be discussed.

### 3.3. Mismatches and Correlation Contributions of k-Words

Table 4 shows the contribution of all tandem repeat words with unit lengths b < 3 to the correlation between IIRs for *k* = 7 and *k* = 11 with zero and with up to one mismatch (for one mismatch, e.g., AAAAGAA is counted as AAAAAAA). Aside from the correlation contributions for all words, the ranking in the word lists is color-coded for no mismatch allowed and given in order to demonstrate not only relative dominance but also absolute influence based on occurrence.

The increase in correlation contribution for almost all of these entire tandem repeats from *k* = 7 to *k* = 11 points to a higher influence of longer sequence stretches in all genomic groups investigated. An interesting insight may be obtained when the ratio of correlation contribution of the sequences with no mismatch is compared with values for those sequences in which up to one mismatch is allowed (Table 6). For *k* = 7, the highest factors between these values, apart from those for polyS and homoS in *Embryophyta* and homoS in *Fungi* and *Animalia* with ratios far above 10, are observed in *Embryophyta* for (TG)n and (AC)n with about 6, for (CA)n and (GT)n at about 4 and polyA/polyT at about 2.75. Factors that are nearly as high are found in *Fungi* with about 4 for (TG)n and (AC)n and about 3.6 for (AG)n and (TC)n. For *k* = 11, the ratio falls far lower to one for all tandem repeats. This indicates a stronger conservation of longer stretches in all genomic sets. This pattern in mutations points towards the characteristic that random mutations are less common and may indicate a function in these tandem repeats.

### 3.4. Deviation Pattern Analysis

If some nucleotide exchanges within functional tandem repeats can be tolerated, we would expect preferences of specific nucleotides within mismatches. Therefore, we checked the mismatches of individual tandem repeat words for such preferences (e.g., if the mismatching nucleotide in AAAAAAA is preferentially C, G or T). Similar methods were developed, published and used in recent publications [8,16,17].

The results within IIRs for *k* = 7 and *k* = 11 are shown in Appendix A and for *Animalia* in [8]. To obtain an even better overview than what is already conducted in these tables by color coding the percentages for the deviations found for each base, a deeper analysis was performed, and a set of rules for nearly all deviations could be identified and some addition general observations could be made. An overview is provided in Table 7.

For some sequence deviations, the values for the percentages in each changed base may come close to a homogeneous distribution among the bases and a valid pattern may not be surely identified. In order to approach this, value sets will be labelled ambiguous if all the percentage values are between 30% and 40% for poly-sequences and between 20% and 30% in addition to a maximum of one value allowed to be 30% (after rounding) for homo-sequences. These conditions are intuitively correct, but they are still arbitrary to a certain degree. As this deviation pattern analysis is performed on sets of sequences and not from a single gene region and also not from a single genome but from multiple genomes, which have their own degrees of heterogeneity within themselves, this approach may still be feasible for finding general characteristics.

The ratios to the extent that observed deviations may be covered by the rules and the ratios that may not be safely attributed by these rules due to ambiguous values are observed in Figure 6.

The only kingdoms in which there are deviation patterns not covered by the rules are those of the heterogeneous sets of *Protista* and *Fungi*. In *Embryophyta* and *Animalia,* all patterns found may be explained by the rules given, and ambiguous value sets appear only in a few cases. If verified for occurrence of ambiguous values, the ambiguous value sets appear to only occur in the homoS sequences in *Embryophyta* and *Animalia*. If checked in all kingdoms, it is mainly in the homoS and homoW sequences. When checked again for strong tendencies in deviations (> 42.5% for at least one value for the occurrence of a base in a STR deviation), this is found for one value mainly in homoR and homoY and for up to even two values in polyS.

This yields about five deviation patterns conserved in all four kingdoms and between IIRs, indicating that these mismatches do not seem to be equivalently distributed for all four bases in all words, which could also again indicate that these sequences have functional aspects.

## 4. Discussion

Recently, STR DNA sequence patterns have been described as conserved within and between introns and intergenic regions in the kingdom of *Animalia* [4,8].

These results have pointed towards a more general, potentially functionally relevant behavior of STR structures so that we extended the investigations on a wider range of genomes here. We again studied multiple genomic regions, including genes, exons, CDS, introns and intergenic regions, of 93 other species beyond *Animalia* consisting of 34 *Embryophyta* (land plants), 30 *Protista* and 29 *Fungi*. We also re-analyzed *Animalia* genomes from [4] and added 18 recently sequenced genomes to that dataset. 

For all investigations, by applying established *k*-mer based (alignment-free) comparison methods and interpretations, it was important to select genomes with fully assembled and annotated versions without biasing the results obtained. These requirements face some challenges when searching for genomes in *Protista* and *Fungi*. For *Protista,* we found enough genomes to representatively cover the phylogenetic tree in this kingdom, even though we would have aimed for more genomes as this kingdom is known to be a very big and contains heterogeneous evolutionary sets of eukaryotes [18]. For *Fungi,* the possible set of genomes to choose from was less diverse. For *Embryophyta* and *Animalia*, nearly all reasonable phylogenetic coverages could be made. The heterogeneity for nearly all results in both the kingdoms of *Protista* and *Fungi*, resulting in the approach via subsets #S1 and #S2, seem to be at least partially linked with a varying G+C content in the genomes. *Embryophyta* and *Animalia* as well as both #S1 subsets are, in contrast, quite homogeneous regarding this parameter, with genomic G+C contents mostly below 45%. Analyzing not entire genomes but special genomic regions, and not only for *k* = 1 (what is equivalent with G+C content) but up to *k* = 11, reveals further patterns depending on *k* and an overall high influence of STRs. This indicates that G+C contents, although surely having an influence, may not be the only reason for these patterns, as A/T and G/C bases seem to be non-randomly arranged in the IIRs.

We found conserved sequence structures in all phylogenetic groups studied. While the patterns observed for *Embryophyta* and *Animalia* were very similar, the same patterns were only found in a subgroup of *Protista* and *Fungi* we named S1. We found positive correlations between coding regions, which are well known to be highly conserved (e.g., exons and CDS) in all groups, but we also found high correlation values between introns and intergenic regions that do not have a well-known shared functions, especially in *Animalia*, *Embryophyta* and the S1 sub-groups. Consistent with former studies, the sequence structures most relevant for these correlations were STRs with repeat unit lengths b ≤2 [6,8]. The observation that the same sequence patterns seem to be responsible for intronic–intronic, intergenic–intergenic and intronic–intergenic correlations indicates phylogenetic conservation and, thus, a preserved function of these structures. Conserved STRs with small repeat unit lengths as functional motifs are the keystones in this hypothesis. Despite all possible mutations and chromosomal rearrangements, the high correlation contributions of these STRs and high correlation values were conserved.

Additionally, not all STRs examined have the same influence on correlation values as their inverse complementary sequence, which is what might be expected in analogy with Chargaff’s second parity rule for single bases [19]. We do find sequence and *k* dependent correlation contributions of STRs, and most of them are in the same range as their inverse complementary sequence and others with their shift sequence in some subsets. As all searches have been performed on one DNA strand, this indicates that for some sequences (homoR/Y, homoK/M) in some regions of the IIRS in at least some genomes of the subsets in *Protista* and *Fungi*, a strand specific unequal distribution is still strong enough to be detectable in this general approach. For *Embryophyta* and *Animalia*, this remains unclear, as the correlation values in this approach over many genomes are too close to each other to exclude possible regions in the IIRs of single genomes. Again, this encourages further research as conserved specific patterns favor a function of these elements. This idea of functional aspects due to sequence dependent behavior receives even higher support by the analysis of single base mismatches in these DNA words.

The deviations in the IIRs do not seem to follow every possible and observable (e.g., A to A would not be seen) change in the bases, but favoring, depending on their own sequence, special transitions. This is consistent with findings of non-random mutations in special sites and *k*-mers [16]. The transitions found are mainly towards A/T and with their own additional rules for homoR and homoY as well as for polyW. These sequence dependent deviations over such large periods of evolution (in the range of two Gya [20]) would point to the directions of general mechanisms that have been favored over evolution and may be due to functional aspects of these sequences. If these deviations are observed as non-random mutations from an original sequence, this may not be performed by one single mechanism such as the known and common enrichment of A/T by transversions or transition mutations from G/C [21,22], as it seems to be sequence dependent and also differing between sequences with the same G/G content (homoR and homoY do not behave as homoK/M).

At this point, we want to point out that polyW stretches and G+C content alone cannot explain the results shown. PolyW repeats were found relevant for all analyses made, but the combined influence of other tandem repeats is higher than the influence of polyW words and we observe differences between tandem repeats sharing the exact same G+C content.

While determining the exact mechanism by which the observed patterns are preserved is outside of the scope of this work, we will discuss reasonable mechanisms in the following. Possible mechanisms explaining shared sequence structures between regions, without indicating functional elements, include the shared origins or constant exchange between those regions. While the formation of a gene from an intergenic region was reported for Mus musculus but appears to be a rather rare process [23], the other direction, namely the emergence of an intergenic region (e.g., a pseudo gene that is not linger-translated or transcribed) from a former gene, is a well-known process [24]. In fact, this was one of the earliest models explaining the existence of non-coding DNA [3,25]. However, this process can only explain conserved sequence structures if mutational rates are low, otherwise the structures would degenerate over time. Therefore, the finding that fast mutating repeat structures are relevant for the conservation makes this model improbable. Another possible mechanism for shared sequence structures is constant mixing or exchange of sequences between the regions. Transposons could be a possible transport medium for this mixing as this family of functional DNA motifs is known to be mobile and shared between IIRs. Recently, transposons were also associated with embryogenesis [26]; therefore, a conserved functional role of these elements would not be surprising. While a detailed analysis of transposon sequences is out of the scope of this article, preliminary results suggest that tandem repeats (>4 bp) are not typical elements of transposons [27,28]. Still, as a result of reverse transcription [29], transposons are known to be flanked by polyA sequences and could, therefore, explain the presence of polyW stretches in IIRs but not of other tandem repeats.

Another known functional sequence motif conserved between IIRs, present in all analyzed organisms, are regulatory elements, such as enhancers and silencers and especially transcription factor binding sites [30]. While the exact sequence structures defining these elements are still not completely understood [31], it was shown that STRs provide unspecific binding sites for at least some transcription factors [32,33]. These findings are of particular interest since unspecific binding of transcription factors regulates their specific binding and, therefore, indirectly affects transcription regulation [34]. Additionally, it was shown that certain repeats of AC are a general feature of enhancers in *Drosophila* [35].

Similarly for transposons, it is known that transcription factor binding sites are often flanked by A/T rich sequence motifs with typical lengths of 3–5 bp [36,37]. These structures cannot explain the presented results since the patterns identified in this work were longer (>6 bp). Another interesting feature of transcription factors is their association with local DNA topology [38]. DNA topology, on the other hand, depends strongly on local dinucleotide compositions [39,40], which is influenced by the presence of dinucleotide repeats identified as conserved in this work. 

Transcription factors are believed to regulate the transcription of distant genes by forming chromatin loops [31]. It seems reasonable that sequence structures favoring such loops support their formation. Such sequence structures are typically A/T rich [41], supporting the hypothesis that the A/T rich repeat structures found to be conserved between introns and intergenic regions influence local bending properties. Since the linear distance between transcription factor and gene also influences loop formation, the bare presence (or length) of a tandem repeat could already influence transcription. In summary, this means that tandem repeats may regulate transcription indirectly by influencing local dimer contents or by favoring certain 3D confirmations.

While these functions could also be implemented by other non-repetitive sequence structures, the fast mutational rates of tandem repeats [42] may allow relatively fast adaption of transcriptional regulation and, therefore, favor repeats.

STRs were also found to colocalize in 3D [43] and, therefore, are potential determinants of global 3D chromatin organization. The very same mechanisms discussed as relevant for local loop formation could also influence the global organization of chromatin and, therefore, explain the mechanism underlying co-localization.

For every mechanism discussed, the selection of different tandem repeats over others observed in this study, especially of A/T rich repeats, could be an important stepping stone for further research. A promising step may be an in-depth analysis of special regions and loci in the individual genomes in the neighborhood of these tandem repeat-based STRs, considering their real individual lengths, distributions and also degree of conservation. This may then provide insights into the possible functions of these element conservations. This may then provide insights into the possible functions of these elements.

## Figures and Tables

**Figure 1 genes-12-01571-f001:**
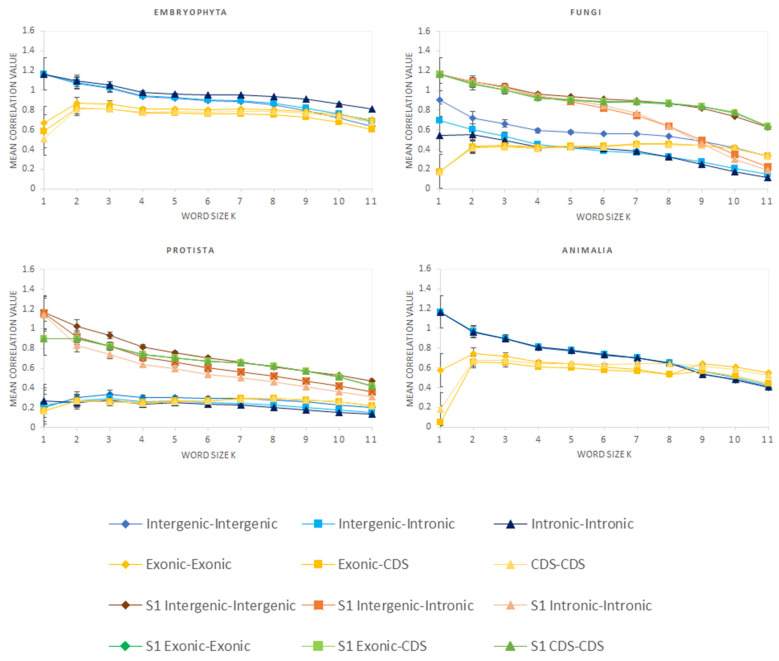
Mean correlation values (difference between genomic data and reference data) over all genomic sets against word length *k* (1 ≤ *k* ≤ 11). The error bars show the combined errors of modelled and real data. Mean correlation values may be above one as negative values may result from the models for the reference data. The curves labeled with “S1” show the results of a selection of a subgroup of higher correlating organisms in *Protista* and in *Fungi*. The *y*-axis is given for values above zero, and connecting lines between points were added for better readability.

**Figure 2 genes-12-01571-f002:**
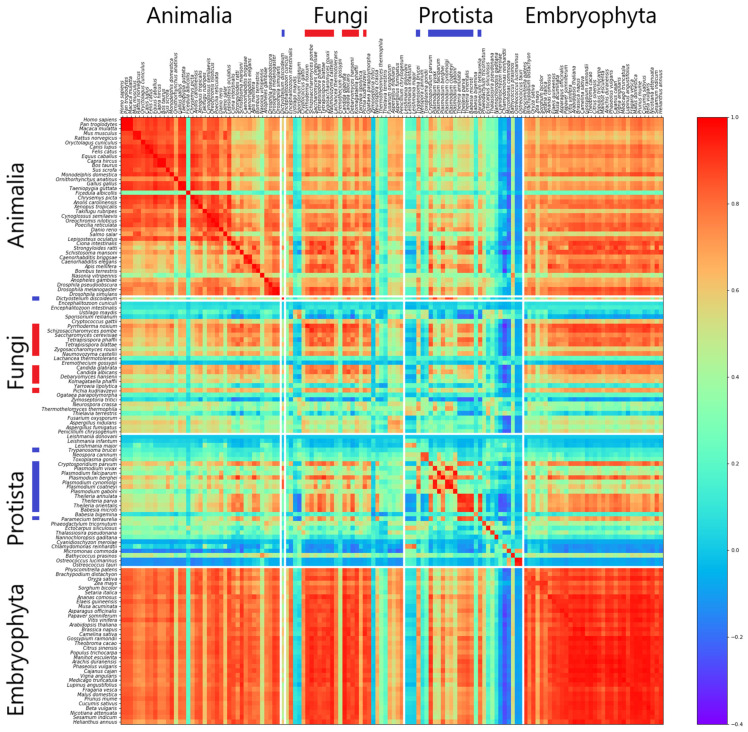
Heatmap of Correlations between individual intronic and intergenic regions (*Animalia*, *Fungi*, *Protista* and *Embryophyta*, *k* = 7). Each row represents the pairwise correlation values (Pearson correlation) of introns for the listed organism, while every column is associated with the intergenic region of an organism. Color scale is limited in heatmap to values above −0.4 for better readability. Genomes belonging to subset 1 in *Protista* are labelled with blue bars, and those of subset 1 in *Fungi* are labelled with red bars on the axes. The protist *Dictyostelium discoideum* of subset #S1 of *Protista* is placed between *Fungi* and *Animalia*, as this very organism is a member of the *Amoebozoa* and has close relations to both kingdoms.

**Figure 3 genes-12-01571-f003:**
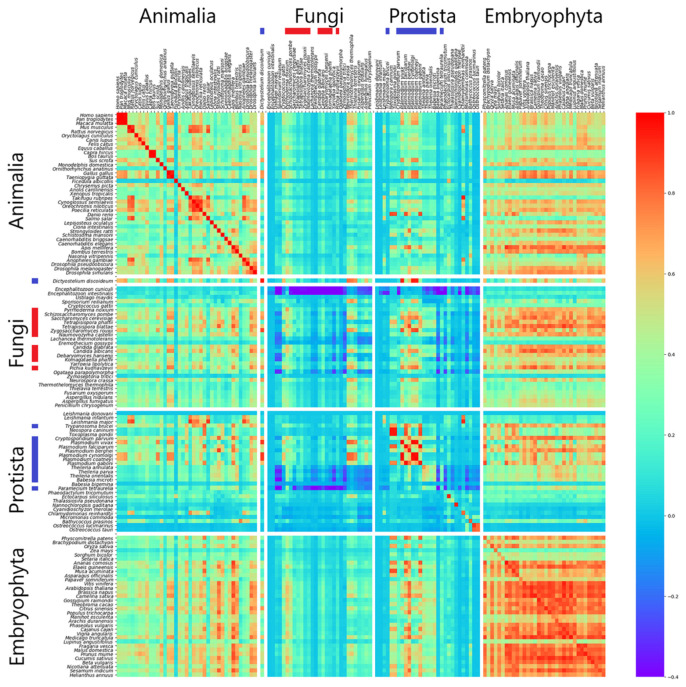
Heatmap showing correlation values between individual intronic and intergenic regions (*Animalia*, *Fungi*, *Protista* and *Embryophyta*, *k* = 11). Each row represents the pairwise correlation values (Pearson correlation) of introns for the listed organism, while every column is associated with the intergenic region of an organism. Color scale is limited in heatmap to values above −0.4 for better readability. Genomes belonging to subset 1 in *Protista* are labelled with blue bars, and those of subset 1 in *Fungi* are labelled with red bars on the axes. The protist *Dictyostelium discoideum* of subset #S1 of *Protista* is placed between *Fungi* and *Animalia*, as this very organism is a member of the *Amoebozoa* and has close relations to both kingdoms.

**Figure 4 genes-12-01571-f004:**
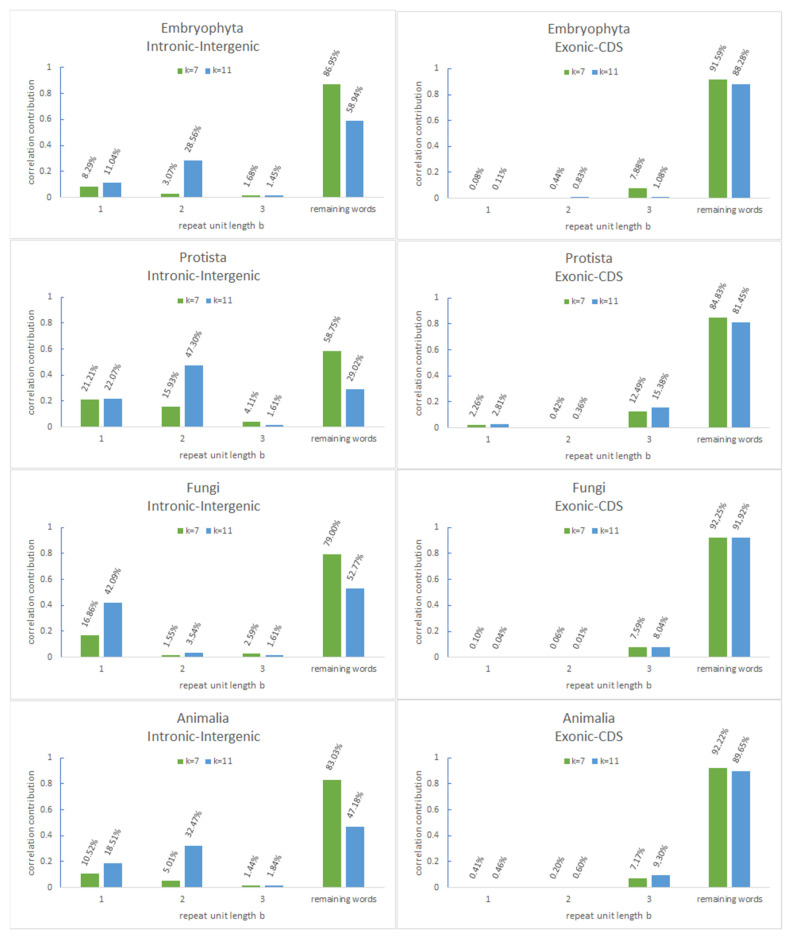
Correlation contribution of different repeat unit lengths (*b* = 1 to *b* = 3) for *Embryophyta*, *Protista*, *Fungi* and *Animalia*. Histograms representing the correlation contributions of tandem repeat *k*-mer words for *k* = 7 and *k* = 11 for intergenic–intronic correlations and for exon–CDS correlations are shown.

**Figure 5 genes-12-01571-f005:**
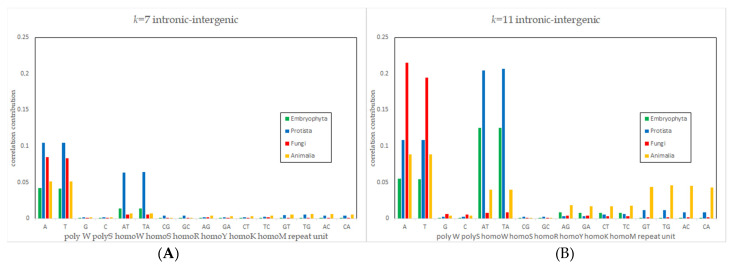
Correlation contributions of tandem repeat words with repeat length *b*=1 and *b*=2 for (**A**) *k* = 7 and (**B**) *k* = 11.

**Figure 6 genes-12-01571-f006:**
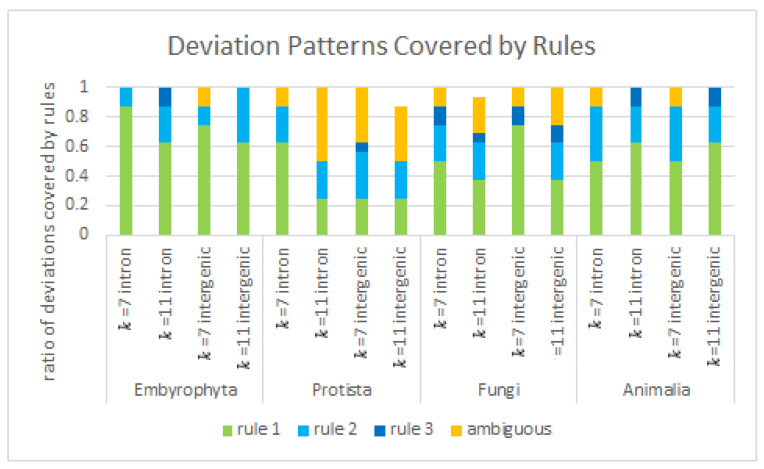
Deviation patterns covered by rules. The coverage ratios by the rules and ambiguous value sets in all four kingdoms for introns and intergenic regions for *k* = 7 and *k* = 11 are given.

**Table 1 genes-12-01571-t001:** Percentage with respect to the genome length for the genomic regions in the three kingdoms.

Kingdom	Intergenic (%)	Genic (%)	Intronic (%)	Exonic (%)	CDS (%)
*Embryophyta*	69.5 ± 12.06	29.94 ± 12.18	16.48 ± 6.43	13.01 ± 8.02	9.03 ± 5.13
*Protista* #S1	31.15 ± 11.05	68.58 ± 11.40	6.71 ± 3.77	61.98 ± 11.62	60.83 ± 11,92
*Protista* #S2	32.81 ± 14.05	66.95 ± 13.74	11.01 ± 14.65	55.53 ± 17.60	51.72 ± 18,89
*Fungi* #S1	29.27 ± 9.43	70.72 ± 9.45	1.70 ± 3.89	68.63 ± 10.49	66.89 ± 10.37
*Fungi* #S2	35.77 ± 14.44	64.21 ± 14.44	4.30 ± 3.24	59.90 ± 16.59	55.57 ± 19.30
*Animalia* (56 genomes)	46.30 ± 11.79	53.70 ± 11.79	42.10 ± 10.97	11.60 ± 16.67	7.10 ± 11.14

**Table 2 genes-12-01571-t002:** Correlation contributions of tandem repeats words (*k* = 7) for different chain seed lengths *b* (1 ≤ *b* ≤ 3). The correlation contributions for correlations between different genomic regions analogous to the data shown in Figure 4 are shown. (E/P/F/A) stands for (*Embryophyta*/*Protista*/*Fungi/Animalia*).

Correlated Regions	Correlation Contribution *b* = 1 (E/P/F/A) in %	Correlation Contribution *b* = 2 (E/P/F/A) in %	Correlation Contribution *b* = 3 (E/P/F/A) in %	Remaining Words (E/P/F/A) in %
**Intergenic–Intergenic**	7.81/22.78/13.31/9.62	3.71/12.54/3.37/4.82	2.03/3.68/2.80/1.54	86.45/61.01/80.53/89.82
**Intergenic–Intronic**	8.29/21.21/16.86/10.52	3.07/15.93/1.55/5.01	1.68/4.11/2.59/1.53	86.95/58.75/79.00/84.02
**Intronic–Intronic**	8.25/19.46/13.61/11.22	2.35/18.68/0.68/5.08	1.37/3.27/1.73/1.36	88.02/58.59/83.98/82.34
**Exonic–Intronic**	4.88/11.64/3.18/8.28	1.76/5.32/0.20/2.81	2.43/6.65/7.51/2.26	90.94/76.40/89.12/86.66
**Exonic–Exonic**	1.47/2.48/0.18/4.02	1.61/0.39/0.05/0.98	6.90/12.51/7.72/5.18	90.02/84.62/92.04/89.82
**Exonic–CDS**	0.08/2.26/0.10/0.41	0.44/0.42/0.06/0.20	7.88/12.49/7.59/7.17	91.59/84.83/92.25/92.22
**CDS–CDS**	0.02/2.05/0.07/0.03	0.14/0.32/0.05/0.09	8.09/12.98/7.88/8.21	91.74/84.66/92.00/92.67

**Table 3 genes-12-01571-t003:** Correlation contributions of tandem repeats words (*k* = 11) for different chain seed lengths *b* (1 ≤ *b* ≤ 3). The correlation contributions for correlations between different genomic regions analogous to the data shown in Figure 4 are shown. (E/P/F/A) stands for (*Embryophyta*/*Protista*/*Fungi/Animalia*).

Correlated Regions	Correlation Contribution *b* = 1 (E/P/F/A) in %	Correlation Contribution *b* = 2 (E/P/F/A) in %	Correlation Contribution *b* = 3 (E/P/F/A) in %	Remaining Words (E/P/F/A) in %
**Intergenic–Intergenic**	8.01/23.84/33.47/17.34	32.17/33.28/6.66/33.40	1.78/2.96/1.81/2.09	58.04/39.92/58.06/47.27
**Intergenic–Intronic**	11.04/22.07/42.09/18.51	28.56/47.30/3.54/32.47	1.45/1.61/1.61/1.84	58.94/29.02/52.77/47.18
**Intronic–Intronic**	13.45/27.16/42.81/20.45	22.94/35.99/2.60/32.35	1.07/2.56/1.25/1.54	62.54/34.30/53.34/45.66
**Exonic–Intronic**	11.20/31.21/11.71/21.31	16.72/9.30/0.89/21.49	2.70/9.71/8.17/2.37	69.38/49.78/79.23/54.83
**Exonic–Exonic**	3.73/3.36/0.29/13.37	12.79/0.46/0.03/8.70	8.66/14.90/8.26/5.70	74.82/81.28/91.42/72.23
**Exonic–CDS**	0.11/2.81/0.04/0.46	0.83/0.36/0.01/0.60	10.78/15.38/8.04/9.30	88.28/81.45/91.92/89.65
**CDS–CDS**	<0.01/2.42/<0.01/0.02	0.04/0.17/<0.01/0.03	10.11/15.11/8.38/9.86	89.84/82.30/91.62/90.09

**Table 4 genes-12-01571-t004:** The correlation contributions for correlations between introns and intergenic regions for tandem repeat *k*-mer words with repeat unit length ≤2 bp for no and up to one allowed mismatch for *k* = 7 (top) and *k* = 11 (bottom) are shown. In rows with no mm—dark green; if in top 4—light green; if in top 16—bright green; if in top 128—orange; if in top 1600—red; if in rest.

	*Embryophyta*		*Protista*		*Fungi*		*Animalia*	
*k*-mer Word	no mm	1 mm	no mm	1 mm	no mm	1 mm	no mm	1 mm
AAAAAAA	4.16%	11.39%	10.47%	15.78%	8.48%	14.91%	5.10%	11.51%
CCCCCCC	<0.01%	0.03%	0.17%	0.36%	0.04%	0.08%	0.17%	0.20%
GGGGGGG	<0.01%	0.03%	0.16%	0.30%	0.04%	0.08%	0.15%	0.18%
TTTTTTT	4.13%	11.34%	10.42%	15.72%	8.30%	14.78%	5.10%	11.55%
ACACACA	0.02%	0.12%	0.39%	0.56%	0.03%	0.13%	0.63%	0.88%
AGAGAGA	0.10%	0.25%	0.18%	0.31%	0.12%	0.44%	0.38%	0.70%
ATATATA	1.35%	2.67%	6.35%	9.26%	0.53%	1.56%	0.68%	1.64%
CACACAC	0.01%	0.04%	0.36%	0.46%	0.02%	0.04%	0.50%	0.57%
CGCGCGC	<0.01%	0.07%	0.35%	0.48%	<0.01%	0.06%	<0.01%	0.08%
CTCTCTC	0.06%	0.10%	0.18%	0.34%	0.09%	0.22%	0.30%	0.43%
GAGAGAG	0.06%	0.10%	0.16%	0.29%	0.08%	0.21%	0.30%	0.43%
GCGCGCG	<0.01%	0.07%	0.35%	0.48%	<0.01%	0.06%	<0.01%	0.08%
GTGTGTG	0.01%	0.04%	0.48%	0.61%	0.02%	0.04%	0.51%	0.58%
TATATAT	1.35%	2.67%	6.40%	9.33%	0.53%	1.56%	0.68%	1.63%
TCTCTCT	0.10%	0.25%	0.20%	0.37%	0.13%	0.45%	0.38%	0.70%
TGTGTGT	0.02%	0.12%	0.54%	0.75%	0.03%	0.12%	0.64%	0.89%
AAAAAAAAAAA	5.48%	7.98%	10.86%	11.85%	21.48%	25.58%	8.84%	10.90%
CCCCCCCCCCC	0.09%	0.10%	0.20%	0.28%	0.52%	0.57%	0.42%	0.45%
GGGGGGGGGGG	0.09%	0.10%	0.20%	0.25%	0.59%	0.64%	0.41%	0.44%
TTTTTTTTTTT	5.39%	7.88%	10.82%	11.83%	19.49%	23.69%	8.83%	10.89%
ACACACACACA	0.09%	0.10%	0.81%	0.85%	0.14%	0.16%	4.48%	4.61%
AGAGAGAGAGA	0.81%	0.84%	0.32%	0.37%	0.38%	0.43%	1.81%	1.91%
ATATATATATA	12.54%	12.87%	20.48%	21.57%	0.75%	0.96%	3.95%	4.20%
CACACACACAC	0.08%	0.09%	0.81%	0.85%	0.14%	0.15%	4.25%	4.36%
CGCGCGCGCGC	<0.01%	<0.01%	0.22%	0.35%	<0.01%	<0.01%	<0.01%	<0.01%
CTCTCTCTCTC	0.75%	0.78%	0.52%	0.57%	0.33%	0.37%	1.68%	1.79%
GAGAGAGAGAG	0.78%	0.81%	0.31%	0.36%	0.37%	0.41%	1.70%	1.79%
GCGCGCGCGCG	<0.01%	<0.01%	0.22%	0.36%	<0.01%	<0.01%	<0.01%	<0.01%
GTGTGTGTGTG	0.08%	0.09%	1.14%	1.19%	0.12%	0.14%	4.32%	4.43%
TATATATATAT	12.54%	12.87%	20.70%	21.73%	0.84%	0.86%	3.95%	4.19%
TCTCTCTCTCT	0.78%	0.81%	0.59%	0.63%	0.34%	0.39%	1.78%	1.89%
TGTGTGTGTGT	0.10%	0.11%	1.18%	1.24%	0.13%	0.15%	4.56%	4.68%

**Table 5 genes-12-01571-t005:** Patterns in value distribution of correlation contributions for *k* = 7 (top) and *k* = 11 (bottom). Descriptions are given on how the subsets of sequences behave. PolyW is either polyA or polyT, polyS is either polyC or polyG and all other sequences are tandem repeats with a repeat unit length of *b* = 2 with W = {A;T}, S = {C;G}, R = {A;G}, Y = {C;T}, K = {G;T} and M = {A;C}. The entries for the value patterns are “inv. compl.” if the close value is in an inverse complementary sequence (for ACACACA this would be TGTGTGT), “shift” if the close value is in a sequence with a shift (for ACACACA this would be CACACAC) and “amb.” if it seems ambiguous between “inv. compl.” and “shift” as all possible values are close to each other. The values of correlation contributions are color-coded, with dark green if both values are above 0.1; light green if both values are above 0.01 up to 0.1; bright green if both values are above 0.001 and up to 0.01; orange if both values are above 0.0001 and up to 0.001; and red if both values are up to 0.0001. * One of the two values slightly in adjacent interval; ** own patterns, polyA with polyG and polyC with polyT; *** variations between inverse complementary and own pattern.

	*Embryophyta*	*Protista*		*Fungi*		*Animalia*
*Tandem repeat subset*		#S1	#S2	#S1	#S2	
	Value Pattern	Value Pattern	Value Pattern	Value Pattern	Value Pattern	Value Pattern
polyW	inv. compl.	inv. compl.	**	inv. compl.	inv. compl.	inv. compl.
polyS	inv. compl.	inv. compl.	**	inv. compl.	inv. compl.	inv. compl.
homoW	inv. compl.=shift	inv. compl.=shift	inv. compl.=shift	inv. compl.=shift	inv. compl.=shift	inv. compl.=shift
homoS	inv. compl.=shift	inv. compl.=shift	inv. compl.=shift	inv. compl.=shift	inv. compl.=shift	inv. compl.=shift
homoR	inv. compl.	amb.	inv. compl.	amb.	inv. compl.	inv. compl.
homoY	inv. compl.	amb.	* inv. compl.	* amb.	inv. compl.	inv. compl.
homoK	inv. compl.	amb.	shift	amb.	inv. compl.	inv. compl.
homoM	inv. compl.	amb.	shift	* amb.	shift	inv. compl.
polyW	inv. compl.	inv. compl.	inv. compl.	inv. compl.	inv. compl.	inv. compl.
polyS	inv. compl.	inv. compl.	inv. compl.	inv. compl.	inv. compl.	inv. compl.
homoW	inv. compl.=shift	inv. compl.=shift	inv. compl.=shift	inv. compl.=shift	inv. compl.=shift	inv. compl.=shift
homoS	inv. compl.=shift	inv. compl.=shift	inv. compl.=shift	inv. compl.=shift	inv. compl.=shift	inv. compl.=shift
homoR	amb.	amb.	*** inv. compl.	inv. compl.	shift	amb.
homoY	amb.	amb.	*** inv. compl.	inv. compl.	shift	amb.
homoK	amb.	amb.	shift	shift	shift	amb.
homoM	amb.	* amb.	shift	shift	shift	amb.

**Table 6 genes-12-01571-t006:** Ratio of correlation contributions of tandem repeat words with up to 1 mismatch and no mismatch allowed. The ratios of correlation contributions of tandem repeat words in Table 4 are shown for correlations between introns and intergenic regions for tandem repeat *k*-mer words with repeat unit length ≤2 bp for no and up to one allowed mismatch for *k* = 7 (top) and *k* = 11 (bottom). If the value is up to 1.25, the row is highlighted dark green; light green if value is above 1.25 up to 1.5; bright green if value is above 1.5 and up to 2; orange if value is above 2 and below 4; and red if value is 4 or above.

	*Embryophyta*	*Protista*	*Fungi*	*Animalia*
*k*-mer Word	Ratio	Ratio	Ratio	Ratio
AAAAAAA	2.74	1.51	1.76	2.26
CCCCCCC	33.39	2.12	2.00	1.18
GGGGGGG	32.00	1.88	2.00	1.20
TTTTTTT	2.75	1.51	1.78	2.26
ACACACA	6.00	1.44	4.33	1.40
AGAGAGA	2.50	1.72	3.67	1.84
ATATATA	1.98	1.46	2.94	2.41
CACACAC	4.00	1.28	2.00	1.14
CGCGCGC	37.27	1.37	129.11	100.44
CTCTCTC	1.67	1.89	2.44	1.43
GAGAGAG	1.67	1.81	2.63	1.43
GCGCGCG	37.30	1.37	118.00	102.16
GTGTGTG	4.00	1.27	2.00	1.14
TATATAT	1.98	1.46	2.94	2.40
TCTCTCT	2.50	1.85	3.46	1.84
TGTGTGT	6.00	1.39	4.00	1.39
AAAAAAAAAAA	1.46	1.09	1.19	1.23
CCCCCCCCCCC	1.11	1.40	1.10	1.07
GGGGGGGGGGG	1.11	1.25	1.08	1.07
TTTTTTTTTTT	1.46	1.09	1.22	1.23
ACACACACACA	1.11	1.05	1.14	1.03
AGAGAGAGAGA	1.04	1.16	1.13	1.06
ATATATATATA	1.03	1.05	1.28	1.06
CACACACACAC	1.13	1.05	1.07	1.03
CGCGCGCGCGC	1.16	1.60	1.71	1.20
CTCTCTCTCTC	1.04	1.10	1.12	1.07
GAGAGAGAGAG	1.04	1.16	1.11	1.05
GCGCGCGCGCG	1.14	1.61	1.84	1.20
GTGTGTGTGTG	1.13	1.04	1.17	1.03
TATATATATAT	1.03	1.05	1.02	1.06
TCTCTCTCTCT	1.04	1.07	1.15	1.06
TGTGTGTGTGT	1.10	1.05	1.15	1.03

**Table 7 genes-12-01571-t007:** Rules and general observations in the deviation patterns of tandem repeats with *b* = 1 and *b* = 2 in introns and intergenic regions. The descriptions of how the subsets of sequences behave are provided. polyW is either polyA or polyT, polyS is either polyC or polyG and all other sequences are tandem repeats with a repeat unit length of b = 2 with W = {A;T}, S = {C;G}, R = {A;G}, Y = {C;T}, K = {G;T} and M = {A;C}. To regenerate deviation patterns observed, rules 1 to 3 have to be applied in this order to sequences with no mismatch.

Number	Sequence Type	Rule	Illustrating Pattern
**1**	Polysequences	First rank for deviations is W (three ranks possible); if S∩R=G in sequence, then into W∩R=A; if S∩Y=C in sequence, then into W∩Y=T	T in polyA, A in polyT T in polyC, A in polyG
	Homosequences	First and second rank for deviations are W (four ranks possible); the same W base as already in the sequence with no mismatch mostly appears on the first rank	A (1.), T (2.) in (AC)_n_ A (1.), T (2.) in (GC)_n_ A (1.), T (2.) in (GT)_n_ T (1.), A (2.) in (TG)_n_
**2**	homoR/homoY and polyW	Second rank is S; if R in sequence, then S∩R=G; if Y in sequence, then S∩Y=C	T (1.), G (2.) in polyA A (1.), C (2.) in polyT A (1.), T (2.) in (AG)_n_ T (1.), C (2.) in (CT)_n_
**3**	polyW	First and second rank changed	G (1.), T (2.) in polyA C (1.), A (2.) in polyT

## Data Availability

All codes and scripts (including visualization) used for this article, as well as a manual, are available online at “http://www.kip.uni-heidelberg.de/biophysik/software” or from an associated GitHub repository (https://github.com/Sievers-A/Oligo. Accessed on 27 September 2021).

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
