# Peer review of "Eukaryotic Genomes Show Strong Evolutionary Conservation of k-mer Composition and Correlation Contributions between Introns and Intergenic Regions"

_genes, 2021, doi:10.3390/genes12101571_

Round 1
Reviewer 1 Report
The aim of the paper seems to be to identify small functional regions of genomes using a reference-free kmer comparison approach across several species based on the relative conservation of these regions. This seems to be an expansion of the methods presented in https://www.mdpi.com/2073-4425/9/10/482/htm , which is an adequate self-reference for this group.
The paper in general is well presented, goes into detail explaining the methodology, describes results in great detail and is overall clear and thorough. The authors claim to have identified small motifs that are likely to be conserved and potentially be functional
There is, however, one thing that I believe is critical but is missing from this paper. In the conclusions, the authors declare:
441: "G/C contents, though surely having an influence,may not be the only reason for these patterns, as A/T and G/C bases seem to be non-random arranged in the IIRs."
However, the authors have not presented any kind of test to establish whether any of these kmers are more or less important than random sequences since there are no random arrangements to compare to. There is no null hypothesis tested.
To be able to make this claim, I submit that the authors should, in fact, test whether they found conservation above that expected from a random sequence or not. Two ways this could be done:
- Given an annotated genome, with intergenic/intron/exon/CDS features, re-arrange the bases found under each given feature to have a random order. If Feature X covers bases 1..100, extract the sequence between bases 1 and 100, re-shuffle them randomly, then replace the sequence in the original genome with this re-shuffled sequence.
- Given a Feature of length l, calculate the frequency of all four bases in the feature, then generate a new sequence of length l using the emission probabilities established for that feature.
The reason for this is easy to explain. Take figure 1 in the paper. Early on, the authors correctly indicate how the comparison between features with a kmer k=1 is pretty much a test of G/C content, so a comparison between Embryophyta genomes and re-shuffled Embryophyta genomes should show the same correlations at k=1 that the pure Embryophyta comparison does. At k=2 there are only 4^2=16 possible kmers, so it is not unlikely that the correlation between the real genomes and the re-shuffled genomes will still be high. by the time k=11, there should be significant divergence between the comparison between real genomes, and the comparison to these "random" genomes.
We know that very short kmers are very rarely unique within a genome, but become rapidly unique given a certain length (see Kurtz et al, https://pubmed.ncbi.nlm.nih.gov/18976482/ , figs 1 and 7), so establishing whether the presence of a kmer in two unrelated regions of the genome, two unrelated features, and even two unrelated genomes has to be done. Specially given how several figures and results show higher correlations with shorter kmers (figs 2 and 3 being an example).
On section 3.2, the authors spend a lot of time analyzing the high relative contribution of b=1 and b=2 repeats. While there is nothing wrong with this, it should be presented as the highest examples of an analysis of higher-than-expected contribution (i.e., a generalized approach to identify outliers), instead of merely looking at the top list without actually establishing "how high is too high".
The fact that the analyses found these is a good thing. We know Poly-A(/T for reverse complement) repeats are present at the tail of every gene, so it's not unexpected to find them dominating the top of the list. 2-mer repeats, often used as microsatellites, are also well known, though their exact role is not as well established. In that sense, it's not entirely surprising to find them, and confirms the method can find kmers from outliers that are more present than they should. But there's no actual test for it.
Also, allowing a mismatch is a questionable approach. The relative contribution of kmers with a mismatch is much higher with the shorter k=7, simply because there are less overall kmers at that length (4^7=16,384). For k=11, the contribution of the Poly-A kmer is much higher because kmers increase in uniqueness with length, so the extra ambiguous base has much less relative impact. Moreover, given a sequence like this:
XXAAAAA...AAAAAXX
Allowing a mismatch would add the kmer XAAAAAA and the kmer AAAAAAX to the contribution of the AAAAAAA kmer, which is very questionable. If a particular mutation appears more often in a tandem repeat than any other mutation, that mutation will appear at the top of the contribution list regardless. Yes, there is redundancy in kmer sets (AXAAA and AAXAAA probably represent the same), but there are better ways to consolidate those possible mutations than grouping kmers with a mismatch (i.e., by establishing first if certain mutation is more prevalent than others). Including k=5 and k=9 in the analysis would likely be more informative than allowing a mismatch.
---
While the above are my criticisms, the overall aims of the paper are good, the analyses thorough, and the presentation is very good. Overall, if the authors include analyses to demonstrate how their findings differ from random kmers, and establish quantifiable significance, the rest of the paper would be solid. I fully expect many if not all of the conclusions to be supported as significant, but it HAS to be done.
Author Response
Dear reviewer,
please find all our answers down below under your suggestions.
You may also see the attachment.
Comments and Suggestions for Authors
The aim of the paper seems to be to identify small functional regions of genomes using a reference-free kmer comparison approach across several species based on the relative conservation of these regions. This seems to be an expansion of the methods presented in https://www.mdpi.com/2073-4425/9/10/482/htm , which is an adequate self-reference for this group.
The paper in general is well presented, goes into detail explaining the methodology, describes results in great detail and is overall clear and thorough. The authors claim to have identified small motifs that are likely to be conserved and potentially be functional
There is, however, one thing that I believe is critical but is missing from this paper. In the conclusions, the authors declare:
441: "G/C contents, though surely having an influence, may not be the only reason for these patterns, as A/T and G/C bases seem to be non-random arranged in the IIRs."
However, the authors have not presented any kind of test to establish whether any of these kmers are more or less important than random sequences since there are no random arrangements to compare to. There is no null hypothesis tested.
To be able to make this claim, I submit that the authors should, in fact, test whether they found conservation above that expected from a random sequence or not. Two ways this could be done:
Given an annotated genome, with intergenic/intron/exon/CDS features, re-arrange the bases found under each given feature to have a random order. If Feature X covers bases 1..100, extract the sequence between bases 1 and 100, re-shuffle them randomly, then replace the sequence in the original genome with this re-shuffled sequence.
Given a Feature of length l, calculate the frequency of all four bases in the feature, then generate a new sequence of length l using the emission probabilities established for that feature.
The reason for this is easy to explain. Take figure 1 in the paper. Early on, the authors correctly indicate how the comparison between features with a kmer k=1 is pretty much a test of G/C content, so a comparison between Embryophyta genomes and re-shuffled Embryophyta genomes should show the same correlations at k=1 that the pure Embryophyta comparison does. At k=2 there are only 4^2=16 possible kmers, so it is not unlikely that the correlation between the real genomes and the re-shuffled genomes will still be high. by the time k=11, there should be significant divergence between the comparison between real genomes, and the comparison to these "random" genomes.
Thank you for the suggestion and the feedback. We understand the method you describe. We think it would effectively produce a reliable reference to overcome the lack of a significance test / null hypothesis for our conclusions. However, with our limited resources, it would be very time consuming (probably weeks or months of additional computations) to perform this method for all genomes and regions analyzed. Additionally, we were not able to calculate all these values, as we, even after several requests, were only given 4 weeks for revision by the journal. Therefore, we had to use a modified version of the approach you suggested. Instead of shuffling the sequences of actual genome regions, we created k-mer spectra using Markov models simulating sequences of 1000 Mbp in size.
We used zero order models as reference for Fig. 1, since at this point we did not want to exclude conserved G+C content as source for the observed correlations.
Later on, we used first order models (which should produce very similar results to the method you suggested) as reference to exclude that the observed tendencies for certain DNA word families (especially for tandem repeats) are a trivial result of similar G+C content.
We recreated Fig. 1 (it now shows the difference of data derived from genomes to the model’s results). We added descriptions of the used Markov models to Methods and added tables with the model results to the appendix (Table S2 and S3).
We know that very short kmers are very rarely unique within a genome, but become rapidly unique given a certain length (see Kurtz et al, https://pubmed.ncbi.nlm.nih.gov/18976482/ , figs 1 and 7), so establishing whether the presence of a kmer in two unrelated regions of the genome, two unrelated features, and even two unrelated genomes has to be done. Specially given how several figures and results show higher correlations with shorter kmers (figs 2 and 3 being an example).
Thank you again for the feedback and also for the source (we added it in the Method section). We think you are right that there is an influence of the word length k to the observed correlation values. To point that out, we added a sentence to Methods 2.2 (93-97). We also checked Results for related statements. Please let us know, if you still see potential for misunderstandings there.
On section 3.2, the authors spend a lot of time analyzing the high relative contribution of b=1 and b=2 repeats. While there is nothing wrong with this, it should be presented as the highest examples of an analysis of higher-than-expected contribution (i.e., a generalized approach to identify outliers), instead of merely looking at the top list without actually establishing "how high is too high".
As already mentioned, we added results from first order Markov models as a reference. We repeated the analysis with the simulated spectra and found that the results observed for non-coding regions were (as expected) significant. The model results can be found in the appendix (Table S3). We also added statements on the significance to 3.2 (252-256).
The fact that the analyses found these is a good thing. We know Poly-A(/T for reverse complement) repeats are present at the tail of every gene, so it's not unexpected to find them dominating the top of the list. 2-mer repeats, often used as microsatellites, are also well known, though their exact role is not as well established. In that sense, it's not entirely surprising to find them, and confirms the method can find kmers from outliers that are more present than they should. But there's no actual test for it.
We agree that some findings were not entirely surprising and that finding known patterns confirms the functionality of our methods. What we consider surprising is the conservation of the presence of 2-mer repeats while, as you said, their exact role/function is still not clear.
Also, we hope that comparing the results to simulated is sufficient a test for significance of the results.
Also, allowing a mismatch is a questionable approach. The relative contribution of kmers with a mismatch is much higher with the shorter k=7, simply because there are less overall kmers at that length (4^7=16,384).
We agree with you about the influence of the word length on these results. We removed statements comparing the correlation contributions between k=7 and k=11 for that reason. See also below.
For k=11, the contribution of the Poly-A kmer is much higher because kmers increase in uniqueness with length, so the extra ambiguous base has much less relative impact.
That´s why we are also using the relative ratios of mismatches/no mismatches in Tab. 6 for arguing and not only absolute values from Tab. 4. But even with absolute values of correlation contributions in Tab. 4 we also find other highly contributing k-words that are not Poly-A as ATn or that e.g. in Animalia ACn is stronger contributing than AGn.
We do not say based on these analyses that we have now proof for functional aspects of these k-mers, but that considering their abundance/correlation contribution they might have an influence on a bigger level than others and that is why we focus on them.
Moreover, given a sequence like this:
XXAAAAA...AAAAAXX
Allowing a mismatch would add the kmer XAAAAAA and the kmer AAAAAAX to the contribution of the AAAAAAA kmer, which is very questionable. If a particular mutation appears more often in a tandem repeat than any other mutation, that mutation will appear at the top of the contribution list regardless. Yes, there is redundancy in kmer sets (AXAAA and AAXAAA probably represent the same), but there are better ways to consolidate those possible mutations than grouping kmers with a mismatch (i.e., by establishing first if certain mutation is more prevalent than others).
All these analyses are done on a sliding window approach/search for k-words. A found pattern „XAAAAAA“ does not mean we have 5´ and 3´ no Poly-A pattern, it might be GGGTXAAAAAACGTGG but also AAAXAAAAAAAAAA. That is why we have given in the supplement an overview of deviation preferences over the whole length of a k-word and not position dependent (it may be for both XAAAA or AXAAA). Finding really mutation prevalences in k-words depending on the position in the k-word would require a different search type. Sliding window is a standard approach in the field and especially if searching in fragmented sequences as introns or intergenic regions over several genomes, this avoids other possible complications as e.g. where to start with a non sliding approach and how valid such a search would be as e.g. intron boundaries may vary by some base pairs and would change the whole reading frame. Searching for real full length of these STRs followed by position dependent mutation prevalence analysis may be done, but we see this beyond scope of this paper.
Including k=5 and k=9 in the analysis would likely be more informative than allowing a mismatch.
According to the results show in Fig. 1, we expect k=5 and k=9 to behave like k=7 and k=11. Nevertheless, it could be interesting to repeat the analysis for other k but we consider this outside the scope of this paper. Also for comparison with [https://www.mdpi.com/2073-4425/9/10/482/htm] we used k=7 and k=11 and the same analysis of missmatches.
---
While the above are my criticisms, the overall aims of the paper are good, the analyses thorough, and the presentation is very good. Overall, if the authors include analyses to demonstrate how their findings differ from random kmers, and establish quantifiable significance, the rest of the paper would be solid. I fully expect many if not all of the conclusions to be supported as significant, but it HAS to be done.
Thank you for your enthusiasm and positive response. We hope that the references and additional model calculations we provided now are sufficient to support the significance of our findings.

Reviewer 2 Report
This article is a follow-on from one published in 2018, using the same methods but applied to a new set of 93 genomes. Unfortunately, it is written assuming that the reader is as familiar with the original article as the authors. More background explanation could be added to the introduction, and likewise more detail in the methods section rather than just referencing the earlier paper.
Author Response
Dear reviewer,
please find all our answers down below under your suggestions.
You may also see the attachment.
Comments and Suggestions for Authors
This article is a follow-on from one published in 2018, using the same methods but applied to a new set of 93 genomes. Unfortunately, it is written assuming that the reader is as familiar with the original article as the authors. More background explanation could be added to the introduction, and likewise more detail in the methods section rather than just referencing the earlier paper.
We do apologize for this inconvenience of referencing the earlier paper for all methods. We do know from other submissions to mdpi, even if we do not have a follow-on paper but just using the same methods and approaches, we will be forced by mdpi to reference the former papers and not allowed to describe much again (they do check for „auto plagiarism“ also in the methods). We already got a check on plagiarism and even k-words as „AAAAAAA“ have been considered as „plagiate“.
We search for k-words with a sliding window approach in different regions as introns, CDS…in this approach here mainly for k=7 and k=11. We get distributions of occurences for each word and we make Pearson correlations of these distributions to check for/quantify divergence/difference. We analyze the impact of each word in this correlation (correlation contribution). So far these are no new techniques and are widely described in the field. It is new to search the intronic and intergenic parts of the genome in so many genomes and also new is that we still find same patterns as in animals what we have done before. We then checked also newly for other patterns found along our research which have not been done before (strand specific occurence of some word patterns and general rules how stable these specific words found are).
We have tried adapting a bit some phrases to make it easier to follow even if the first paper is not known.

Reviewer 3 Report
General comments.
I read with interest the draft manuscript “Eukaryotic Genomes show a Strong Evolutionary Conservation of k-mer Composition and Correlation Contributions between Introns and Intergenic Regions” of Liane Sauer et al. It is an interesting manuscript on the analysis of conserved k-mers in genomes of species from different branches of the tree of life. I found the methodological approach sound for the analysis of multiple genomes for conserved k-mers and of some relevance due to the inclusion of all sorts of organisms in the analysis. Although this analysis, with a more complete “coverage” of multiple branches of the tree of life, gives interest to the manuscript, as a first general comment, the manuscript globally fails showing originality since an important part of the objectives, methodology, results and discussion overlap with the previous research article “Conservation of k-mer Composition and Correlation Contribution between Introns and Intergenic Regions of Animalia Genomes” of Sievers et al. (2018, doi:10.3390/genes9100482). On the other hand, relevant differences are found on groups of Protists and Fungi, which has been reported in the results, however is not extensively commented in the discussion. I reckon that perhaps the problem may come on the way that the manuscript is written and organized which is my second general comment. Although the text is written in a good and correct English with very few typos, I also found the text difficult to read and follow. A third general comment regards the inclusion of a new Animalia dataset in all the analysis. Given the eukaryotic scope of the analysis, it is necessary to update the genome set of Animalia as a control to check if same patterns of k-mer are present in the updated dataset, or perhaps there is a taxonomical group that also deviates from the previously observed patterns as in Protists and Fungi. Therefore, I recommend that the manuscript text should be extensively revised and the text rewritten, improving the result presentation, avoiding the use of ambiguities or generalizations on the text and highlighting and emphasizing better the differences on what has been previously reported and what is new about the updated analysis of a more phylogenetically complete database of Animal genomes with the reported analysis of the kingdoms studied.
Specific comments:
The next specific comments are regarding the text, the writing and the format issues that apply also to other parts of the whole document. So I advise the authors to review not only the specified comments but also other parts of the new text (in my suggested rewriting of the manuscript) due to the inclusion of a larger updated set for Animalia and for all the modifications that will be done to improve the clarity of the text, tables and figures.
P2L55: The Animalia database should be updated beyond the 39 genomes previously analysed. In case of showing in the results only a subset, specify how the selection was done. This should be done in case there is one phylogenetic branch of the tree of life in Animalia that might strongly deviate from the normal distribution.
P2L84: it should be “k-mer analysis” instead of “k-mer-analysis”
P2L92: it should be “scripts” instead of “scrips”
P2L94: the software written for the manuscript should be available in a public repository independent from one institution web site in order to guarantee that is permanently accessible.
P3L96: Table 1 is too large to be shown in the manuscript. It should be put in supplementary data if it is kept in the same format, otherwise the data should be presented and summarized differently.
P4L101: For comparison reasons, analyses of an updated Animalia genome dataset should be included in the result part. If they show exactly the same, as a positive control and comparison with the other groups, and if change, to identify possible groups or Animalia genomes that strongly deviates from the mean. This also applies to the other results in the manuscript.
P4L106: This grouping should also be mentioned in materials and methods and the nomenclature should be improved to avoid similarities with the supplementary data nomenclature.
P4L121: Supplementary data, tables and figures should be provided in PDF format in order to avoid display differences depending on the software used for opening docx documents.
P5L122: The definition of the subsets 1 and 2 for Protists and Fungi should be included also in materials and methods.
P5L123: The heatmaps need to be better ordered to show the #S1 and #S2 groups and other similarities and differences within the kingdoms analysed.
P5L130: The nomenclature of genomic elements should follow the same pattern in the text and table and figures for the elements analysed. It should be used for instance the adjectives “Intergenic”, “genic”, “intronic”, “exonic” and not interchange adjectives with nouns (for instance in P5L136, Table 2, and Figure 1)
P5L127-142: The authors mention that the correlations are highest for intron(ic)-intron(ic), intron(ic)-intergenic but in the graph they use the words “coding”, intergenic, introns. This lack of consistency in the nomenclature and mixture of adjectives with nouns in plural and singular makes the text hard to read and follow in the manuscript. I reckon the effort of the authors for better organizing the data by using similar colors and including text in Figure 1A, 1B and 1C, however the aforementioned lack of consistency make the results hard to read. In the same Figure 1, it is difficult to differentiate, light-blue from light light-blue in Fig 1B and 1C, perhaps the used of dashed or dotted lines for #S1 subset improves the clarity of this figure. Instead of “light curves“ the term “thin” and “thick” may be more adequate. Finally, the text mentions “deep” slope which is a subjective term, perhaps “more pronounced” slope is more correct. The use of this “subjective” adjective here and also of other adjectives and adverbs in other parts of the text is one of the reasons why I mentioned the “ambiguities” as a systematic problem in the manuscript text.
P5L143: Table 2 is before Figure 1 while in the text the order is the opposite, Figure 1 is mentioned first that Table2.
P5L150: All scientific names should be in italics, in this case for “Vitis vinifera”
P5L155: Following the criterium of consistency, the nomenclature of “G/C” is correct for describing the content of guanines plus cytosines, however it is most frequently written as “GC content” or “G+C content” but not “G/C content”. The readability may be also improved if one of these more common forms is used.
P6 L167: For comparative reasons and also as a control, Figure 1 should include the results from the updated Animalia dataset.
P7L175: Figure 2 and all the figures should be in a vector format not only to improve the quality but also to allow the text search of their contents. In addition, it should be included clearly which criteria was used to show only k=7 and k=11 and why other k-mer lengths were discarded and how the figures change using these values. This also applies to other heatmaps and graphs.
P8L191: Figure 3 is incomplete in the pdf file available for reviewing.
P9L204: Ambiguity in “intron and intergenic correlation”, is it intergenic-intro(nic) or all the corelations between intronic and intergenic regions (intronic-intronic, intronic-intergenic, intergenic-intergenic). Same for other parts of the manuscript text.
P9L213: Lack of spaces in “k=11”, it should be “k = 11”
P10L216: Table 3A and 3B should be merged into one and be simplified or better organized since they both have the same structure.
P10L239: SGC perhaps should be SG/C according to the nomenclature selected by the authors.
P11L256: Figure 5 yellow color should be replaced by perhaps the “gold” color used in other figures. In addition, since the use of other IUPAC nomenclatures (W, S, R, Y K and M) in the results, Figure 5 needs to include the analysis of these “groupings”.
Author Response
Dear reviewer,
please find all our answers down below under your suggestions.
You may also see the attachment.
Comments and Suggestions for Authors
General comments.
I read with interest the draft manuscript “Eukaryotic Genomes show a Strong Evolutionary Conservation of k-mer Composition and Correlation Contributions between Introns and Intergenic Regions” of Liane Sauer et al. It is an interesting manuscript on the analysis of conserved k-mers in genomes of species from different branches of the tree of life. I found the methodological approach sound for the analysis of multiple genomes for conserved k-mers and of some relevance due to the inclusion of all sorts of organisms in the analysis. Although this analysis, with a more complete “coverage” of multiple branches of the tree of life, gives interest to the manuscript, as a first general comment, the manuscript globally fails showing originality since an important part of the objectives, methodology, results and discussion overlap with the previous research article “Conservation of k-mer Composition and Correlation Contribution between Introns and Intergenic Regions of Animalia Genomes” of Sievers et al. (2018, doi:10.3390/genes9100482). On the other hand, relevant differences are found on groups of Protists and Fungi, which has been reported in the results, however is not extensively commented in the discussion.
Thank you for the feedback. We agree that we mostly confirmed the existence of the same patterns already observed in Animalia for Embryophyta. We already see this as a major result, especially since the patterns were not present in every branch of the tree of life. Nevertheless, we rephrased large parts of Discussion in the hope of highlighting all important findings, including the differences found for Protista and Fungi (see 467-673).
I reckon that perhaps the problem may come on the way that the manuscript is written and organized which is my second general comment. Although the text is written in a good and correct English with very few typos, I also found the text difficult to read and follow.
To handle your feedback with the required care, we checked the whole text and rephrased many passages. We put much effort into taking care of good grammar and readability and hope that we removed possible misunderstandings or uncertainties such that it is now less difficult to read and follow the text.
A third general comment regards the inclusion of a new Animalia dataset in all the analysis. Given the eukaryotic scope of the analysis, it is necessary to update the genome set of Animalia as a control to check if same patterns of k-mer are present in the updated dataset, or perhaps there is a taxonomical group that also deviates from the previously observed patterns as in Protists and Fungi. Therefore, I recommend that the manuscript text should be extensively revised and the text rewritten, improving the result presentation, avoiding the use of ambiguities or generalizations on the text and highlighting and emphasizing better the differences on what has been previously reported and what is new about the updated analysis of a more phylogenetically complete database of Animal genomes with the reported analysis of the kingdoms studied.
Thank you for that suggestion. We agree that many Animalia genomes were sequenced and assembled since our first publication on Animalia. Therefore, we decided to add another 18 genomes and repeated major parts of the analysis with the extended data set. We added the results to the appendix and updated Fig. 1 including Animalia. Since we could not find significant differences for the extended data set and since we wanted to focus on other kingdoms in this publication, we decided not to repeat the correlation decomposition for the extended set. Also, the correlation decomposition is a computationally very costly and would probably take weeks if not months to calculate using our limited resources. Additionally, we were not able to calculate all these values, as we, even after several requests, were only given 4 weeks for revision by the journal.
Specific comments:
The next specific comments are regarding the text, the writing and the format issues that apply also to other parts of the whole document. So I advise the authors to review not only the specified comments but also other parts of the new text (in my suggested rewriting of the manuscript) due to the inclusion of a larger updated set for Animalia and for all the modifications that will be done to improve the clarity of the text, tables and figures.
As already mentioned, we rewrote considerable contents of the text. We also included results for Animalia.
P2L55: The Animalia database should be updated beyond the 39 genomes previously analysed. In case of showing in the results only a subset, specify how the selection was done. This should be done in case there is one phylogenetic branch of the tree of life in Animalia that might strongly deviate from the normal distribution.
It is true, new fully assembled genomes with annotations have been published and are available since the last publication from us on this topic. When we have been choosing them we already have tried covering as many phylogenetic branches in Animalia as we could find. And yes, we had only a few genomes of Aves and Reptilia and none in Mollusca, also some „bridging“ genomes were not available. But, we have right now already more genomes in Animalia than in every other phylogenetic kingdom (34 Embryophyta (land plants), 30 Protista and 29 Fungi genomes). We are already also on a coverage level far better in depth and homogeneity than for most other phylogenetic kingdoms given. As reference and for testing and comparison, this should be enough on a solid base. For nearly all results in the paper this would also not have many new implications, if we would integrate all of these genomes, as we are mainly writing about patterns of and in STRs in introns and intergenic regions.
But, we have now searched for these new genomes and found about 18 new ones being able to cover many of these gaps. As written above we have made some analyses on the new set.
P2L84: it should be “k-mer analysis” instead of “k-mer-analysis”
has been changed
P2L92: it should be “scripts” instead of “scrips”
has been changed
P2L94: the software written for the manuscript should be available in a public repository independent from one institution web site in order to guarantee that is permanently accessible.
We additionally uploaded the software to GIT repository and added a link to the software availability statement.
P3L96: Table 1 is too large to be shown in the manuscript. It should be put in supplementary data if it is kept in the same format, otherwise the data should be presented and summarized differently.
has been changed
P4L101: For comparison reasons, analyses of an updated Animalia genome dataset should be included in the result part. If they show exactly the same, as a positive control and comparison with the other groups, and if change, to identify possible groups or Animalia genomes that strongly deviates from the mean. This also applies to the other results in the manuscript.
View above
P4L106: This grouping should also be mentioned in materials and methods and the nomenclature should be improved to avoid similarities with the supplementary data nomenclature.
We rephrased the sentence to clarify that we analyzed CDS, exons, introns and intergenic regions and that we used the same definitions as given in the former publication (111-112)
P4L121: Supplementary data, tables and figures should be provided in PDF format in order to avoid display differences depending on the software used for opening docx documents.
We agree and it will be done by mdpi, when published.
P5L122: The definition of the subsets 1 and 2 for Protists and Fungi should be included also in materials and methods.
This has been a descriptive approach induced by the results of data analysis. It is not a method ab initio. We have rephrased the relevant explanations in the Results.
P5L123: The heatmaps need to be better ordered to show the #S1 and #S2 groups
has been changed
and other similarities and differences within the kingdoms analysed.
Covered by the rewriting (see above)
P5L130: The nomenclature of genomic elements should follow the same pattern in the text and table and figures for the elements analysed. It should be used for instance the adjectives “Intergenic”, “genic”, “intronic”, “exonic” and not interchange adjectives with nouns (for instance in P5L136, Table 2, and Figure 1)
has been changed wherever it was appropriate.
It is problematic as there is no substantive from intergenic, the next is intergenic region what often is too long in tables or figures. If then also „CDS“ comes in where we do not see a suitable adjective.
P5L127-142: The authors mention that the correlations are highest for intron(ic)-intron(ic), intron(ic)-intergenic but in the graph they use the words “coding”, intergenic, introns. This lack of consistency in the nomenclature and mixture of adjectives with nouns in plural and singular makes the text hard to read and follow in the manuscript. I reckon the effort of the authors for better organizing the data by using similar colors and including text in Figure 1A, 1B and 1C, however the aforementioned lack of consistency make the results hard to read. In the same Figure 1, it is difficult to differentiate, light-blue from light light-blue in Fig 1B and 1C, perhaps the used of dashed or dotted lines for #S1 subset improves the clarity of this figure. Instead of “light curves“ the term “thin” and “thick” may be more adequate. Finally, the text mentions “deep” slope which is a subjective term, perhaps “more pronounced” slope is more correct. The use of this “subjective” adjective here and also of other adjectives and adverbs in other parts of the text is one of the reasons why I mentioned the “ambiguities” as a systematic problem in the manuscript text.
Besides the rewriting, we changed the layout, description and labeling of Fig. 1 to improve readability.
P5L143: Table 2 is before Figure 1 while in the text the order is the opposite, Figure 1 is mentioned first that Table2.
has been changed, now Figure 1 before Table 2 (now Table 1)
P5L150: All scientific names should be in italics, in this case for “Vitis vinifera”
has been changed
P5L155: Following the criterium of consistency, the nomenclature of “G/C” is correct for describing the content of guanines plus cytosines, however it is most frequently written as “GC content” or “G+C content” but not “G/C content”. The readability may be also improved if one of these more common forms is used.
has been changed into G+C content
P6 L167: For comparative reasons and also as a control, Figure 1 should include the results from the updated Animalia dataset.
As suggested, we added Animalia to Fig. 1. And in this figure we also use the new Animalia set.
P7L175: Figure 2 and all the figures should be in a vector format not only to improve the quality but also to allow the text search of their contents. In addition, it should be included clearly which criteria was used to show only k=7 and k=11 and why other k-mer lengths were discarded and how the figures change using these values. This also applies to other heatmaps and graphs.
We agree in general, while unfortunately not every software allows vector formats as output format and it would be complicated and time consuming to export, convert and recreate all images. We will definitely consider this for future publications. Since mdpi allows non-vector formats we hope it is convenient for you if we do not provide the images in vector format.
P8L191: Figure 3 is incomplete in the pdf file available for reviewing.
We can assure that the file we uploaded was complete. This may be an issue of the mdpi website or related to problems with the file size.
P9L204: Ambiguity in “intron and intergenic correlation”, is it intergenic-intro(nic) or all the corelations between intronic and intergenic regions (intronic-intronic, intronic-intergenic, intergenic-intergenic). Same for other parts of the manuscript text.
has been changed
P9L213: Lack of spaces in “k=11”, it should be “k = 11”
has been changed
P10L216: Table 3A and 3B should be merged into one and be simplified or better organized since they both have the same structure.
We tried to combine the tables as suggested but since we would have to add an additional column indicating the word length k which would affect the readability of the tables. Therefore, we decided to keep two tables.
P10L239: SGC perhaps should be SG/C according to the nomenclature selected by the authors.
has been changed
P11L256: Figure 5 yellow color should be replaced by perhaps the “gold” color used in other figures. In addition, since the use of other IUPAC nomenclatures (W, S, R, Y K and M) in the results, Figure 5 needs to include the analysis of these “groupings”.
has been changed

Round 2
Reviewer 1 Report
This second version of the manuscript presents substantial changes to the methods, and I'm glad to see the authors taking the reviewers' comments on board.
Having gone over the revised manuscript, the methodology is quite improved with much tighter and carefully defined claims, with good support from the data. The decision to use Markov Models to generate the random sequences instead of more computationally costly methods is understandable, and having reviewed the results, I believe it to be adequate, too.
As it is, I'm happy to recommend this manuscript be accepted.
Reviewer 3 Report
The authors have thoroughly revised the text and figures, and appropriately answered the concerns raised by this reviewer.
Minimal text corrections:
L699: Mus musculus should be italics
L773: it says "3d confirmations" and should be "3D conformations"
L777: it says "in 3D" and should be "in chromosomal 3D folding"